# Complex Interactions in Regulation of Haematopoiesis—An Unexplored Iron Mine

**DOI:** 10.3390/genes12081270

**Published:** 2021-08-20

**Authors:** Ranita De, Kulkarni Uday Prakash, Eunice S. Edison

**Affiliations:** Department of Haematology, Christian Medical College, Vellore 632004, India; ranita.de@cmcvellore.ac.in (R.D.); kulkarni.uday@cmcvellore.ac.in (K.U.P.)

**Keywords:** iron, homeostasis, haematopoietic stem cells, haematopoietic lineages, iron deficiency and overloading

## Abstract

Iron is one of the most abundant metals on earth and is vital for the growth and survival of life forms. It is crucial for the functioning of plants and animals as it is an integral component of the photosynthetic apparatus and innumerable proteins and enzymes. It plays a pivotal role in haematopoiesis and affects the development and differentiation of different haematopoietic lineages, apart from its obvious necessity in erythropoiesis. A large amount of iron stores in humans is diverted towards the latter process, as iron is an indispensable component of haemoglobin. This review summarises the important players of iron metabolism and homeostasis that have been discovered in recent years and highlights the overall significance of iron in haematopoiesis. Its role in maintenance of haematopoietic stem cells, influence on differentiation of varied haematopoietic lineages and consequences of iron deficiency/overloading on development and maturation of different groups of haematopoietic cells have been discussed.

## 1. Introduction

Iron belonging to transition elements is one of the most abundant metals on earth and is vital for the functioning of life forms. It plays an important role in the physiology of plants and animals, including human beings as it is an essential component of several proteins and enzymes involved in numerous metabolic pathways. An important aspect of cellular biology affected by this vital element is ‘haematopoiesis’.

This process occurs chiefly in the bone marrow in adult humans and is responsible for the continued replenishment of haematopoietic cells of different lineages through progressive differentiation of ‘haematopoietic stem cells’ or HSCs [1]. Iron, through specific chemical processes such as the Fenton reaction catalyses the formation of toxic reactive oxygen species or ‘ROS’, which in turn affects HSC development and differentiation [2,3]. A haematopoietic lineage in which iron has undisputed significance is the erythroid lineage, as a component of haemoglobin of erythrocytes. Different hierarchical models have indicated that erythrocytes and megakaryocytes share a common origin. Significance of iron levels in affecting platelet counts has been well documented in the past, with iron deficiency being closely linked with increased platelet counts i.e., thrombocytosis [4]. Several independent studies have reported the significance of iron in influencing different aspects of the development of other haematopoietic lineages including macrophage polarisation, neutrophil biology and lymphocyte maturation among others. The present review summarises the key players involved in iron metabolism and homeostasis identified till date and presents an overview of the role played by iron in affecting haematopoiesis. Consequences of iron deficiency and overloading on development of different haematopoietic lineages have also been discussed.

## 2. Overview of Iron Homeostasis and Metabolism

Iron metabolism in human beings is a complex process, which involves the interaction of several proteins. A clear picture of how this vital nutrient is absorbed, stored and metabolised intracellularly has emerged with the discovery of many key players involved in different stages of the physiological process. In humans, iron is chiefly absorbed through enterocytes of the small intestine, with the help of a transporter protein called the ‘divalent metal transporter 1’ (DMT 1 or SCL11A2) [5]. Iron in the ferric state is absorbed with the aid of a membrane-bound ferrireductase called duodenal cytochrome *b* (DCYTB) which reduces it to its ferrous state [6]. Upon entry into cells, a minor fraction of the intracellular iron exists in weakly chelated form as the ‘Labile iron pool’ or LIP [7]. Most of it is oxidised to its ferric state by ferritin and stored in a non-reactive, non-toxic form in cells [8]. The sole iron exporter protein identified till date, ferroportin (FPN or SLC40A1) transports ferrous iron outside cells [9], with the aid of multicopper ferroxidases such as hephaestin (HEPH) [10] (primarily expressed in the intestine) and its paralog in other tissues; ceruloplasmin (CP) [11]. Iron does not exist in its free form in the serum and is predominantly bound to a globular protein called transferrin (TF) in its ferric state [12]. Iron-loaded TF is internalised by most cells through the transferrin receptor (TfR) [13] into intracellular acidic organelles; endosomes. This facilitates the release of the ferric iron, which is acted upon by ferrireductases (STEAP proteins) and transported to the cytosol by DMT 1, thus completing the cycle. Under normal physiological conditions, the saturation range of TF ranges from 20–35%. When iron levels are in excess, typically observed in disorders associated with iron overloading such as thalassemia and haemochromatosis, iron exists in a free form in the plasma. This ‘Non-transferrin-bound iron’ i.e., NTBI [14] is usually reduced at the cell surface, before being transported to the cytosol by DMT 1.

Iron is a key component of a multitude of enzymes and proteins, and excess levels may lead to the generation of highly reactive, toxic agents such as ROS. Several homeostatic mechanisms have been identified at the cellular and systemic level, which maintain optimum iron levels. Principal means of cellular regulation of iron homeostasis is mediated by the binding of iron regulatory proteins (IRP1 and 2) to certain target sequences called ‘iron response elements’ (IREs) in the untranslated regions of mRNAs of genes involved in iron metabolism [15]. As a result of this interaction, target genes are either up -regulated/down-regulated to facilitate adequate supply of iron to cells during conditions of iron deficiency or overload. At the systemic level, iron levels are regulated by a small peptide hormone synthesised and secreted by the liver, known as hepcidin [16]. This protein controls the absorption of iron by enterocytes of the small intestine, by binding to the iron exporter protein, FPN and promoting its degradation in an iron-replete state, thereby preventing further iron uptake and maintaining homeostasis.

Apart from these classic players, recent studies have unearthed several new proteins which are known to be involved in different aspects of iron metabolism, shedding more light on this complex process. A novel family of transmembrane zinc transporters have recently gained importance for their role in mediating iron absorption. These proteins of the ‘solute carrier family 39 (SLC39) or ZRT/IRT-like protein (ZIP) family’ are involved in the uptake of ferrous iron in a broad spectrum of organisms including plants, yeast, nematodes and humans [17]. ZIP14 plays an important role in the uptake of non-transferrin-bound iron (NTBI) and enables cytoplasmic transport of iron derived from transferrin, similar to DMT 1. Similar to the canonical DCYTB, a ferrireductase known as ‘prion protein’ or PRNP, acts in association with ZIP14 to aid in iron uptake [18]. Another member of the ZIP family, ZIP8 also participates in NTBI during iron overloading and is closely related to ZIP14. However, its tissue expression profile differs from the former [19]. Despite the presence of multiple proteins to aid in the absorption of iron, they seem to have non redundant functions as ZIP8 and 14 operate at physiological pH, while DMT 1 functions at an acidic pH to transfer ferrous iron from endosomes to the cytosol [20]. Thus, different families of transporter proteins seem to have evolved so that the cellular requirements for iron can be met in diverse situations. These novel iron transporters play a crucial role in mediating iron uptake during haematopoiesis as mice lacking ZIP8 are known to suffer from perinatal lethality due to the disruption of haematopoiesis and organogenesis of spleen, liver, kidney and lung [21]. Studies investigating the route of delivery of iron to the non-toxic intracellular storehouse, ferritin have led to the discovery of a novel class of cytosolic iron chaperone proteins. ‘Polycytosine-binding proteins’ or PCBPs are a class of RNA and DNA-binding proteins which bind to cytosine-rich tracts of nucleic acids and transfer iron to ferritin by protein–protein interactions [22]. While two members of this family, PCBP1 and 2 have been studied in detail, others such as PCBP3 and 4 are expressed at lower levels and their precise role as iron chaperones remains vague, although PCBP3 is known to bind to ferritin [23]. These proteins may also be significant in maintaining iron homeostasis, as PCBP2 binds to FPN [24] but its effects on iron export mediated by the latter necessitates further studies. Some recent insights into the fate of intracellular iron sequestered in ferritin has elaborated on the phenomenon of ‘ferritinophagy’ during which it is incorporated into an autophagosome and eventually transferred to lysosomes where ferritin degradation and iron release occurs. The pathway of delivery of ferritin to lysosomes remained elusive, until the discovery of a cargo receptor protein, ‘nuclear receptor coactivator protein (NCOA4)’, which is involved in the transport of ferritin to autophagosomes to facilitate its further degradation [25].

As greater number of genes and proteins involved in the complex physiological processes of iron metabolism emerge, it will help us to investigate their significance in developmental regulation and disease pathology.

An overview of key players, identified in regulating different aspects of iron metabolism is depicted in Figure 1 (Adapted from [26]).

## 3. Iron in Haematopoiesis—A Vital Micronutrient

‘Haematopoiesis’ or the process of formation of blood cells of different lineages is responsible for their continued replenishment and produces billions of blood cells every day. While it is regulated by a plethora of proteins, enzymes and regulatory molecules, iron is an important metal influencing different branches of haematopoiesis. Variation in iron levels, resulting from its deficiency or excess may adversely affect the haematopoietic process and interfere with the normal development and maturation of different haematopoietic lineages.

### 3.1. Role of Iron in Haematopoietic Stem Cell Maintenance and Differentiation

An intricate balance exists between proliferation, differentiation and quiescence in haematopoietic stem cells or HSCs, which is essential for their maturation into different haematopoietic lineages as well as maintenance of self-renewing population of HSCs. Most of them are in a quiescent phase (cell cycle arrested in G0 phase), while relatively lesser number exist in an activated phase (period of cell growth and division) [27]. As they are present in hypoxic niches of the bone marrow, they are highly sensitive to oxidative stress. Iron in its freely available form as LIP may contribute to the production of ROS through chemical processes such as the Fenton and Haber–Weiss reaction [28]. Reports have indicated that various states of HSCs require different ROS levels for their survival and maintenance. While in the quiescent phase, located near hypoxic osteoblastic cells of the bone marrow, HSCs survive in the presence of low ROS levels. As they become activated and possess decreased self-renewal ability, they migrate towards sinusoids of the marrow. This movement occurs along a gradient of increased ROS concentration [29]. A probable reason why hypoxic conditions are favourable during the quiescent phase of HSCs may be protection against DNA damage due to the low oxygen environment, which in turn maintains their self-renewal capacity. This oxygen poor niche could also support the switch from glycolysis to oxidative metabolism during activation and lineage commitment of HSCs [30].

However, very low levels of ROS affect differentiation of HSCs, thereby impairing their capacity to repopulate [31]. As levels of ROS increase, HSCs differentiate to form short-term repopulating cells which eventually develop into the myeloid lineage [32]. Thus, HSC biology is influenced by surrounding ROS levels in the HSC niche, which in turn is intimately linked to intracellular iron levels.

#### Effects of Iron Deficiency and Overloading on HSC Biology

A significant source of different types of ROS is the mitochondrial respiratory chain, occurring during the terminal stages of cellular aerobic respiration. The former includes a series of protein complexes present in the inner mitochondrial membrane, through which electrons are transferred from electron donors to the terminal electron acceptor, i.e., molecular oxygen [33]. Complex III is one of the protein complexes involved in this chain and it contains ‘Rieske iron-sulphur proteins’ (RISP) which participate in electron transfer. Iron is a component of the RISP and loss of the latter in HSCs of foetal mice, affects their differentiation, leading to anaemia and prenatal death. While, inactivation of the RISP in HSCs of adult mice affects their quiescence causing pancytopenia and lethality [34]. Thus, iron seems to be essential for the proper functioning of certain components of the mitochondrial respiratory chain through which it indirectly influences maintenance, differentiation of HSCs and determines the life span of foetal as well as adult mice models.

One of the common routes of iron uptake by most cells occurs through the transferrin receptor (TfR) and this pathway seems to be crucial for the differentiation of HSCs. Recent studies involving *Tfr1* knockout mice were unable to adequately absorb iron and displayed impaired proliferation and differentiation of haematopoietic precursor cells which affected development of most haematopoietic lineages, resulting in postnatal lethality [35]. These observations indicate the significance of optimal iron levels in regulating early HSC differentiation and proliferation. While iron is essential for regulating HSC biology, excess amounts of free iron in the form of labile plasma iron (LPI) [36] or ‘labile cellular iron’ (LCI) is harmful to them [37]. Excess intracellular iron generates different types of ROS, which are highly reactive and may cause injury to HSCs. They also decrease the number of haematopoietic progenitor populations, as indicated by studies involving iron-overloaded mice models [38]. Recent in vitro studies using murine embryonic cell lines treated with exogenous iron, reported increased ROS production, which caused injury to both HSCs as well as differentiated haematopoietic cells [39]. Very high levels of ROS caused by several factors including iron overloading and chronic inflammation may also lead to HSC death by apoptosis [40]. The significance of optimal ROS levels in HSC maintenance is indicated by common malignancies resulting from aberrant intracellular ROS levels, such as in myelodysplastic syndromes (MDS) [41], acute myeloid leukaemia (AML) [42] and other tumours. Apart from increased ROS levels, MDS patients with iron overloading have elevated expression of a mitogen-activated protein kinase (MAPK), P38 in hematopoietic stem/progenitor cells (HSPCs). This may promote progression of MDS to AML in this group of patients [42]. It is interesting to note that patients suffering from hereditary hemochromatosis do not have significantly altered haematopoiesis, despite being subjected to chronic iron overloading [43]. An inherent protection rendered to HSCs within their bone marrow niche, which may not be fully replicated in an in vitro scenario may be responsible for this phenomenon [44]. A multitude of proteins involved in iron homeostasis play an important role in regulating intracellular iron levels. Deficiency of one such protein, the F-box and leucine-rich repeat protein 5 (FBXL5) in HSCs causes iron overloading and results in reduced cell numbers, impaired self-renewal and stem cell exhaustion. HSCs from patients suffering from MDS have reduced expression of FBXL5 which may contribute to disease pathology [44]. Thus, iron overloading seems to adversely affect HSC biology and has several pathological consequences.

### 3.2. Iron in Maintenance of Mesenchymal Stem Cells

The possible role of iron levels in development and maintenance of mesenchymal stem cells (MSCs) has generated interest in recent years. The latter were first described as bone-marrow-derived clonogenic cells that are capable of giving rise to differentiated cells such as osteoblasts, adipocytes and chondrocytes [45]. Over the years, different populations of bone marrow stromal cells have been identified, having similar properties as the originally isolated MSCs [46]. After being isolated from the bone marrow [47], they have been subsequently extracted from different types of adult tissues. Bone-marrow-derived MSCs (BM-MSCs) influence lineage commitment of haematopoietic stem cells, their mobilisation and exit from the bone marrow thereby influencing organisation of the haematopoietic niche and haematopoiesis. Significance of iron in MSC maintenance is indicated by the fact that those damaged by iron-induced toxicity are unable to support haematopoiesis effectively, as observed in diseases such as β thalassaemia and MDS [48].

Iron can enter MSCs through transferrin-dependent and independent pathways. Excess iron levels induce changes in cell signalling pathways and alter different characteristics of MSCs. ROS produced during iron overloading may hinder proliferation of MSCs by arresting the cell cycle at G0/G1 phase [48]. Alternatively, iron overloading may also increase the expression of cell cycle proteins such as cyclin A and E, which stimulates entry of BM-MSCs into the S phase and activates downstream signalling pathways such as the MAP kinase (MAPK) pathway, promoting their proliferation. However, it has been reported that excess iron inhibits osteoblastic commitment of BM-MSCs and matrix calcification, thereby affecting their differentiation into osteoblasts. This may explain the severe bone alterations observed in iron overloading syndromes [49]. Negative effects of excess iron on BM-MSCs can be reversed by exogenous application of antioxidants such as melatonin [50] and iron-binding glycoprotein lactoferrin [51], further indicating the importance of optimal iron concentration in maintenance of MSCs. Iron chelators help in combating oxidative stress, which may be induced by iron excess conditions in BM-MSCs. Increased expression of transcription factors such as HIF-1α in response to iron-deficient conditions may increase viability of BM-MSCs in these situations [52]. Rescue from the effects of iron overloading may also result from an indirect decrease in ROS levels by reducing levels of its activators such as NOX4 in iron-treated BM-MSCs and iron-loaded mice, as well as by inhibiting p38 MAPK signalling pathway in iron-treated umbilical cord MSCs by iron chelators. Certain intracellular signalling pathways, responsible for mitochondrial fragmentation in iron-loaded BM-MSCs from MDS patients may be attenuated by iron chelators, providing another example of the harmful effects of iron excess conditions on MSC physiology [48].

### 3.3. Role of Iron in Erythropoiesis

An intricate network of transcription factors influences differentiation of HSCs into erythroid lineage. GATA1 belonging to a family of DNA-binding proteins with a zinc (Zn) finger motif [53] is an important transcription factor regulating erythropoiesis. Iron in its ferrous state is essential for the formation of an active binding complex of GATA1 with its target DNA sequences [54], thus indicating the significance of iron in early transcriptional control of erythropoiesis. Iron also plays a crucial role in the regulation of erythropoiesis, as a structural component of erythrocytes. Its high requirements during erythropoiesis are met through recycling of senescent erythrocytes in the liver and spleen. After being phagocytosed by macrophages, the resulting phagosomes fuse with lysosomes to form structures called erythrophagolysosomes [55]. Degradation of erythrocytes occurs, and haemoglobin breaks down to release free heme. This in turn is transported to the cytosol via a heme transporter protein, HRG1 [56]. After reaching the cytosol, heme breaks down in the presence of an enzyme known as heme oxygenase 1 (HMOX1) [57] and free iron is released. This may be either exported outside macrophages through FPN or stored intracellularly as ferritin. Like most other cells, erythrocytes take up iron primarily through the transferrin receptor. Significance of this route of iron uptake may be inferred from occurrence of iron deficiency anaemia (IDA) in patients or mice with atransferrinemia [55]. Some studies have indicated an alternate path of iron uptake in in vitro conditions, where ferritin released from macrophages enters erythroblasts through pinocytosis in a transferrin-independent manner [58].

During certain situations such as early developmental stages of humans and pregnancy, an increase in rate of erythropoiesis necessitates greater demand for iron [30]. This is met by a phenomenon known as ‘stress erythropoiesis’, where an increased production of the hormone ‘erythropoietin’ (EPO) by kidneys induces the expression of another recently discovered hormone, ‘erythroferrone’ (ERFE) by erythroid progenitors in the bone marrow. The latter inhibits hepcidin secretion by sequestering bone morphogenetic protein (BMP) ligands, especially BMP6 which impairs the BMP-SMAD signalling pathway; in turn affecting synthesis of hepcidin [59].

Another recently identified protein, the growth differentiation factor 15 (GDF15), produced by the late stage erythroid precursors has also been proposed to mediate hepcidin suppression during ineffective erythropoiesis. Its precise function in diseases linked to iron deficiency requires further studies [60]. Low iron levels or oxygen deprivation also drives erythrocyte production in an EPO responsive manner by inhibiting hydroxylation activity of an enzyme known as ‘prolyl hydroxylase’ or PHD [61]. This results in the stabilisation of the oxygen-dependent α subunit (HIF-1α, HIF-2α) of a transcription factor known as ‘Hypoxia inducible factor’ (HIF), which stimulates erythropoiesis in response to EPO secreted by kidneys [62]. Thus, iron levels are intimately linked to erythropoietic output in several ways.

#### Effects of Iron Deficiency and Overloading on Erythropoiesis

One of the most common causes of anaemia worldwide is iron deficiency, which is responsible for 50% of all anaemias. Several causes including insufficient iron intake, decreased absorption, or blood loss has been attributed to iron deficiency which in turn affects erythropoiesis and results in hypochromic, microcytic erythrocytes characteristic of iron deficiency anaemia (IDA) [63]. A key regulator of systemic iron homeostasis, the peptide hormone hepcidin is intricately linked to erythrocyte production and its deregulation has been linked with different types of anaemias. While iron deficiency is associated with low hepcidin levels, iron overloading conditions are characterised by increased hepcidin expression which inhibits further intestinal iron absorption. However, abnormally high synthesis of hepcidin leading to iron restricted erythropoiesis is observed in anaemias such as iron refractory iron deficiency anaemia (IRIDA), anaemia of acute and chronic inflammatory disorders despite the presence of low iron stores and low serum iron respectively [64]. Iron overloading due to reduced hepcidin synthesis, resulting from chronic stress erythropoiesis/ineffective erythropoiesis is also a trademark of disorders such as β thalassemia [65].

In iron-replete conditions, erythropoiesis may be modulated by a transferrin receptor called TfR2, which is a homolog of TfR1. TfR2 forms a complex with the EPO receptor in erythroblasts, senses high iron concentrations in the liver and probably down-regulates the number of EPO receptors to limit erythropoiesis. Thus, expression of TfR2 in erythroblasts regulates erythropoiesis, while in hepatocytes it influences hepcidin expression and iron uptake, as per erythropoietic needs. Recently its expression has been reported in osteoclasts and osteoblasts, which indicates a novel role for this iron sensor in modulation of erythropoiesis, iron homeostasis as well as bone turnover [59]. Defective TfR2 in mice and humans has been associated with iron overloading [66]. Consequences of iron loading on erythropoiesis and early haematopoietic development have been investigated through in vitro studies. For instance, bone marrow cells isolated from patients with iron overloading show increased apoptosis of HSCs and diminished differentiation of erythroblasts due to generation of ROS. This leads to activation of the p38 MAPK pathway which apart from decreasing self-renewal capacity of HSCs, activates p53 which activates pro apoptotic proteins causing death of HSCs [67]. Studies investigating the effects of iron overloading in mice have shown a reduction in the number of immature haematopoietic cells and clonogenic capacity of HSPCs through an increase in ROS levels by the NOX4/ROS/P38 MAPK signalling pathway. Treatment of HSPCs with iron chelators reversed the above-mentioned effects [38]. Thus, exposure to high iron levels may affect erythropoiesis and haematopoietic development via ROS-induced oxidative injury, indicating the significance of iron homeostasis for the normal progression of haematopoiesis.

Iron deficiency may also adversely affect erythropoiesis, as indicated by in vitro studies where in iron deprivation inhibited the maturation of early erythroid progenitors by loss of their response to the hormone EPO. Underlying mechanism involves inactivation of the mitochondrial aconitase; multifunctional iron-sulphur cluster proteins. Supplementation by isocitrate reversed the anaemia progression in iron deprived mice [59]. Most of the data investigating significance of iron overloading and deficiency on erythropoiesis and haematopoiesis have been obtained from in vitro studies and murine models; hence these results may not be accurately replicated in humans. Clinical trials are required for understanding the effects of varying iron levels on haematopoiesis and lineage differentiation in humans [29]. Investigation of the sensitivity of different erythropoietic stages to varying iron concentrations could identify stages of erythropoiesis that are most sensitive to iron overloading and deficiency. This may pave way for future medical interventions for overcoming erythropoietic anomalies associated with aberrant iron levels.

### 3.4. Intricate Balance between Iron Levels and Megakaryopoiesis—An Incompletely Understood Phenomenon

Iron levels and megakaryopoiesis seem to be closely associated as several past reports have indicated that some patients suffering from mild to moderate IDA exhibit thrombocytosis [68]. Few occurrences of thrombocytopenia/decreased platelet counts in patients suffering from severe IDA have also been reported [69]. It is important to understand the megakaryocytic process and external factors affecting platelet biogenesis to understand the role of iron levels in its regulation. Megakaryopoiesis involves the progressive maturation of megakaryocytes to thrombocytes/platelets, which enables coagulation of blood from damaged blood vessels during an injury. As per the classic hierarchical models of megakaryopoiesis, a common ‘multipotent progenitor’ or MPP bifurcates into the common myeloid (CMP) and lymphoid progenitors (CLP) [70]. The former splits into the granulocyte–macrophage (GM) and a common megakaryocyte-erythroid progenitor (MEP). This in turn differentiates into unipotent megakaryocytic progenitors, giving rise to megakaryocytes and platelets [71] apart from giving rise to erythroid cells. Subsequent studies have hypothesised multiple origins for megakaryocytes. Similarities in surface markers, transcription factors and signalling pathways between HSCs and megakaryocytic progenitors have pointed towards a direct origin of megakaryocytes from HSCs. This new hierarchical model proposes that certain populations of HSCs are capable of producing megakaryocytes, bypassing the CMP and MEP stages [72]. Despite the current existence of a multitude of pathways explaining the origin of megakaryocytes and platelets, the significance of varying iron levels on megakaryopoiesis cannot be ignored. IDA may be a significant factor responsible for reactive thrombocytosis in many anaemic patients, along with increased megakaryocytic differentiation, greater states of ploidy and increased number of megakaryocytic progenitors [73], though precise mechanism(s) responsible for these remain vague. Isolated cases have indicated a close association of iron overloading with pancytopenia, including low platelet counts which have been corrected with aggressive iron chelation therapy [74]. Excess iron levels have also been linked to dysfunctional platelets in patients receiving chronic blood transfusions [75].

An early theory developed by Karpatkin et al. known as the ‘two-compartment model’, states the significance of iron in maintenance of an optimal platelet count either directly or/indirectly. This may explain IDA-associated thrombocytosis. They also suggested an additional role of iron in inducing maximum platelet biogenesis through the production of ‘megathrombocytes’. Severe iron deficiency (ID) effects the latter functional role of iron, leading to thrombocytopenia in patients affected with severe IDA [76]. However, the specific level of ID which activates alternating aspects of this model is yet to be known. Another popular theory known as the ‘stem cell steal’ phenomenon may explain the significance of iron in thrombopoiesis [77,78]. Different models of haematopoietic differentiation have suggested a common progenitor stem cell for erythroid and platelet precursors. During conditions of ID leading to anaemia, stimulation of the hormone EPO may induce megakaryopoiesis, leading to increased platelet production. In iron-replete conditions however, the common progenitor cell diverges towards the erythroid lineage and thrombopoiesis is hindered, leading to transient thrombocytopenia observed in a minority of patients with severe IDA following iron therapy [79]. Other possible mechanisms by which iron may influence platelet biogenesis have been studied in iron-deficient animal models and in vitro experiments. Platelets obtained from iron-deficient rats show altered characteristics such as increased number, size and aggregation [80]. These changes are unlikely to be mediated by common thrombopoietic cytokines such as IL-6, IL-11, thrombopoietin (TPO) as their levels in ID groups did not vary significantly from controls. The role of EPO in influencing platelet count remains controversial. While recombinant human EPO increases platelet count in patients suffering from renal failure [81], in vitro studies have indicated the stimulatory effect of EPO on megakaryocytic proliferation, in combination with other megakaryocytic cytokines such as TPO [82]. However, EPO alone supports very little megakaryopoiesis [83], probably due to the loss of the EPO receptor on maturing megakaryocytic progenitors [84]. Moreover, platelet count show poor correlation with EPO levels, the latter may increase in events apart from thrombocytosis [83,85]. Thus, iron deficiency itself may be responsible for increased platelet numbers in some IDA patients rather than a stimulatory cytokine [80].

A search for target genes through which iron deficiency may stimulate megakaryopoiesis revealed hypoxia-inducible factor (HIF) and vascular endothelial growth factor (VEGF) as potential targets. Genes involved in these pathways were induced in iron deficiency. Increased expression of HIF-2α, its targets including VEGF-R1, VEGF-R2 and their cognate ligand-VEGF-A was observed in megakaryocytic cell lines propagated in iron-deficient conditions [73]. It was proposed that iron deficiency may induce the expression of HIF-2α from megakaryocytes, which in turn may enhance megakaryopoiesis by sustained expression of VEGF-A. At the same time the development of the alternative erythroid lineage from megakaryocyte-erythroid progenitor (MEP) is suppressed as haemoglobin synthesis does not occur in iron-deficient conditions [73]. This also explains thrombocytosis observed in IDA patients. Recent investigations inspecting molecular mechanisms linking iron deficiency and megakaryopoiesis indicated that MEPs isolated from iron-deficient mice exhibited decreased proliferation and increased bias towards the megakaryocytic lineage compared to wild type MEPs. Gene expression studies showed an increased expression of genes involved in metabolic and genes encoding VEGF in MEPs obtained from iron-deficient mice. These data agreed with the findings observed in iron-deficient primary human MEPs. Signal transduction analyses further indicated that human and mice iron-deficient MEPs had reduced levels of phosphorylated ERK proteins. Thus, iron deficiency seems to be a key player that modulates cellular metabolism and intracellular signalling pathways in megakaryocytic progenitor cells and drives them towards a megakaryocytic fate [86].

### 3.5. Significance of Iron in Myeloid Lineage

Apart from the erythroid and megakaryocytic lineages, certain members of the myeloid lineage are closely linked with iron. Phagocytic cells such as macrophages occupy the centre stage in iron homeostasis and aid in its recycling from senescent erythrocytes. Tissue macrophages are in turn dependent on iron for their functional development [87]. Iron deficiency has been associated with certain morphological anomalies in neutrophils [88]. It also decreases apoptosis [89] in them, thus emerging as a factor affecting apoptotic response in neutrophils.

#### 3.5.1. Iron and Macrophage—A Two Way Story

Macrophages in the bone marrow play a crucial role in the developmental regulation of early erythroid progenitors by providing various signals. As erythroid differentiation proceeds, structures called ‘erythroblastic islands’ are formed where a central macrophage interacts with up to 30 erythroblasts. Macrophages mediate iron supply and regulate proliferation, differentiation of erythroblasts in these islands. They also play an important role in regulating erythroblast maturation to the reticulocyte stage by phagocytosing nuclei extruded by these cells [90]. A class of erythrophagocytic macrophages present in the red pulp of spleen, bone marrow and liver participate in iron homeostasis by phagocytosing senescent erythrocytes. The major amount of iron required for heme biosynthesis in erythroblasts is provided by this route. Following engulfment by these macrophages, heme present in erythrocytes is free from haemoglobin and transported to the cytoplasm [91]. Accumulation of heme activates a heme-responsive gene; heme oxygenase-1 (HO-1) which liberates iron and generates biliverdin and carbon monoxide (CO) [92]. The residual iron present in macrophages may be excreted out via FPN or stored intracellularly as ferritin [91].

As macrophages play a crucial role in regulating heme iron metabolism, the latter stimulates differentiation and functional development of tissue-resident macrophages. Monocytes in blood differentiate into erythrophagocytic macrophages upon stimulation by a heme-responsive transcription factor known as SPI-C (Spi-1/PU.1 related) [93]. Macrophages have been classically categorised into the pro inflammatory M1 subtype and anti-inflammatory M2 subtype, based on their polarisation responses. The latter category express a high degree of iron cycling due to increased expression of FPN and low ferritin levels, making M2 macrophages suitable for tissue repair and angiogenesis [94]. Thus, intracellular iron levels are also closely associated with the elicitation of appropriate phagocytic and regenerative responses from macrophages, through certain stable and reversible transcriptional programs known as ‘polarisation’, in response to environmental cues [95]. In M2 macrophages, increased breakdown of heme occurs in presence of HO-1, which generates CO and promotes secretion of interleukin 10 (IL-10) in response to the stimulation of macrophages by external microbial components such as lipopolysaccharides. This cascade of events acts as a positive feedback loop to induce further activation of HO-1, continuing macrophage polarisation towards tissue regeneration in presence of heme. M1 macrophages on the other hand, store iron due to increased expression of ferritin and decreased expression of FPN. They also possess bactericidal activity and produce cytokines, rendering them pro inflammatory [95].

A recent function attributed to self-renewing tissue-resident macrophages is their ability to maintain homeostasis and act like ‘ferrostats’ that sense and respond to local tissue iron needs, and regulate the tissue microenvironment [94]. Specialised populations of adipose tissue macrophages, which have the ability to uptake excess iron, thereby protecting adipocytes from iron overloading present an example of this functional role played by macrophages [96]. Such iron-cycling macrophages are being identified in other tissues and organs as well. Accumulation of iron in macrophages may have clinical implications. Recent evidence suggests that build-up of iron in macrophages due to deficiency of Fpn in mice models dramatically increased the progress of atherosclerosis. Possible causative factors may include elevated levels of ROS, aggravated systemic inflammation and altered plaque-lipid composition. As these symptoms were relieved by iron chelation, control of intracellular iron levels in macrophages by systemic iron chelation/dietary restriction may serve as a potential novel intervention strategy in patients suffering from atherosclerosis [97].

#### 3.5.2. Role of Iron in Regulation of Neutrophil Biology

Iron has an essential role in controlling cellular apoptosis, specifically the intrinsic apoptotic pathway. The latter involves release of cytochrome c from mitochondria to initiate the apoptotic cascade. Iron is an important component of cytochromes [98]. Studies investigating the significance of iron deficiency in affecting apoptosis in neutrophils have indicated that children with IDA had lower apoptotic rates than healthy controls, which increased upon iron supplementation. Decreased neutrophil apoptosis in iron-deficient conditions may induce a pro-inflammatory state, leading to autoimmune disorders or malignancies in the future [89]. Contradictory reports have indicated occurrences of neutropenia in some patients affected with severe IDA. A recent case of a middle aged woman with severe IDA exhibiting neutropenia, which recovered following intravenous iron therapy presents yet another example of the significance of iron deficiency in affecting neutrophil biology [99]. Observation of neutropenia in some severe IDA patients may also explain the increased risk of infections reported in them [100].

Optimum levels of iron are indispensable for eliciting the oxidative response of neutrophils, through generation of ROS. However, iron overloading in chronically transfused patients severely affects neutrophil functions. It is interesting to note that in patients suffering from hereditary haemochromatosis (HH) and HH mouse models, absence of hepcidin expression prevents the accumulation of intracellular iron and protects neutrophils from its toxic effects. In these patients, systemic iron overloading was instead associated with neutrophil priming and increased their oxidative burst, thereby enhancing phagocytosis [100]. Iron may also be important for the maintenance of neutrophil morphology. Reports of neutrophil hypersegmentation (NH) i.e., the presence of five or more lobes in ≥5% of neutrophils have been reported in some IDA patients. This abnormal phenotype is thought to arise due to impaired DNA synthesis, because of folate deficiency or an inability of neutrophils to utilise folate. Several studies have indicated that iron deficiency may be an underlying cause of NH as iron supplementation reverts neutrophils back to their normal morphology. However, as folate deficiency is a common occurrence in cases of IDA, the specific cause of NH needs to be determined before proceeding with treatment [88]. Anti-microbial functions of neutrophils may be attributed to an iron-dependent metalloprotein known as myeloperoxidase (MPO) present in their azurophilic granules. The latter is a haemoprotein and the alternate redox states of an iron atom present in its heme prosthetic group is critical for ROS production and anti-microbial activity [101].

## 4. Iron and Immunity

A close relationship exists between iron and the immune system, which emerged as an evolutionary strategy developed by an organism to withhold iron from invading pathogens, acting as a defence mechanism. Thus, proteins involved in iron trafficking are regulated by cytokines and acute phase proteins. Conversely iron promotes ROS-dependent killing of bacterial pathogens, stimulates proliferation and effector functions of T lymphocytes and it is indispensable for cell division, cytokine production as a component of heme and numerous Fe-S containing enzymes [101]. Transcription factors such as NF-kB are activated by iron in hepatic macrophages, which in turn regulates multiple genes involved in innate immunity and inflammation [102].

### Significance of Iron in Development of the Lymphoid Lineage

Iron plays an important role in the metabolic and redox reactions during the proliferation and effector functions of T lymphocytes. They undergo rapid expansion, while developing in the thymus and during an immune response to invading pathogens. Hence to fulfil their requirements for iron, an up-regulation of TfR/CD71 on T lymphocytes is one of the earliest cues during their development [101]. A principal route of iron uptake during development of T and B lymphocytes occurs through TfR. Mice models in which embryos carry a homozygous deletion of the gene encoding Tfr (designated as *Tfr-/-)* fail to undergo differentiation of their T lymphocytes which are arrested at the triple negative state (CD3-CD4-CD8-). Mice models in which embryos carry a homozygous deletion of Tfr on a recombination activating 2 (*Rag2*) null background (*Rag2-/-)* showed impaired B lymphocyte differentiation and only some chimeric mice produced IgM antibodies on surfaces of B lymphocytes. Other antibody classes were not expressed, indicating the significance of Tfr during early lymphocyte maturation. These mice were reported to survive until embryonic day 12.5, following which they died due to anaemia. However, most of their other embryonic tissues showed normal development [103,104]. Thus, transferrin mediated iron uptake seems to be essential for lymphopoiesis and erythropoiesis. Iron uptake through TfR1, to support extensive cell proliferation during lymphocyte development may explain the significance of this metal in maintenance of lymphopoiesis on the whole [104]. However, greater sensitivity of T lymphocytes as compared to B lymphocytes as indicated by the presence of immature thymocytes, towards iron deficiency remains a mystery.

Studies investigating the mechanism of iron uptake by lymphocytes have yielded contradictory results. Some patients with congenital atransferrinemia have normal B and T lymphocytes, suggesting that iron required for lymphocyte development may be met by transferrin-independent pathways [105]. In support of this hypothesis, it has been reported that uptake of non-transferrin-bound iron (NTBI) by T lymphocytes increases expression of ferritin and ferroportin, through the IRE-IRP regulatory mechanism. T lymphocytes play a protective role in preventing iron overloading of organs such as liver and spleen in mice, by acting as a ‘circulating storage component’ of NTBI [106]. Evidence favouring transferrin-dependent iron uptake of lymphocytes include some early reports of inhibition of proliferation of B and T lymphocytes by use of monoclonal antibodies against TfR1 [107]. Similar results have been reported in mice models carrying a homozygous deletion of *Tfr*, which develop poorly differentiated T and B lymphocytes [104].

Although iron is essential in developmental regulation of lymphocytes, it has an inhibitory effect on certain aspects of their maturation. For instance, iron negatively affects rearrangement of the immunoglobulin heavy chain genes and consequent class switching in B lymphocytes. Immunoglobulin (Ig) class switch DNA recombination (CSR) and somatic hypermutation (SHM) are crucial processes for the production of mature antibodies of different isotypes (or classes) by B lymphocytes [108]. These processes are mediated by an enzyme known as activation-induced cytidine deaminase (AID) [109]. Ferrous iron suppresses the process of CSR, thus affecting Ig class switching and leading to reduced number of different antibodies such as IgG1, IgG3 and IgA [108]. Recent reports have hinted at the significance of intracellular iron levels in epigenetic programming of the promoter region of cyclin E1 to control B-cell activation, proliferation and antibody responses. The proposed mechanism is ‘Histone 3 lysine 9 (H3K9)’demethylation, which is repressed during iron deficiency. This was confirmed by reduced antibody responses in iron-deficient mice and humans with IDA, on antigen exposure [110]. Storage of intracellular iron in a non-toxic form in ferritin protects cellular components from ROS-induced oxidative stress. However, mutations in genes encoding ferritin adversely affect lymphocyte development. Deletion in the ferritin heavy chain (*Fth*) gene results in reduced lymphocyte population, without affecting other haematopoietic lineages [111]. Presence of functional FTH and its ferroxidase activity are required for scavenging excess labile iron, thereby inhibiting cellular toxicity [112]. *Fth* deletion leads to an increase in labile iron which may generate ROS, causing apoptosis and reduced lymphocyte numbers. This may not be the only factor contributing to cell death. Lymphocytes possessing a high LIP show increased mitochondrial depolarisation due to proton pump uncoupling in the mitochondrial membrane, leading to superoxide production [111]. This in turn causes cellular damage due to generation of oxidative stress. Thus iron, affects humoral as well as cellular branches of adaptive immunity in multiple ways and its homeostasis ensures normal development of the lymphoid lineage.

Recent reports indicate that anaemic children have altered T-cell counts, compared to controls. While CD4+ T cells were significantly reduced in the IDA group, CD8+ T-cell counts were higher indicating the importance of early diagnosis and treatment of asymptomatic cases of IDA before cellular immunity is further compromised [113]. Restoration of normal lymphocyte counts upon intravenous iron therapy in an IDA patient, signifies the effect of iron deficiency on white blood cells which has not yet been investigated in detail [99]. Excess iron levels also affect lymphocyte counts. Patients suffering from diseases associated with iron overloading such as hereditary haemochromatosis have decreased CD8+ T cells, while CD4+ T-cell counts are unaffected. The former exhibits a more pronounced effector phenotype, suggesting a close link between increased intracellular iron levels and enhanced proliferation of T cells. In patients suffering from β thalassemia major, an alternate situation is seen where CD4+ T cells decline and CD8+ T-cell counts increase [101]. Iron deposition is a hallmark of several autoimmune diseases, including neuroinflammatory diseases such as multiple sclerosis (MS) where lymphocytes are crucial mediators of abnormal immune responses. In affected individuals, iron deposition has been linked to neurodegeneration, inflammation resulting from oxidative injury and is detected in some lesions through techniques such as magnetic resonance imaging [114]. Another common example of an association of elevated iron levels with abnormal cellular expansion is observed in cancer cells, which have evolved a multitude of mechanisms to evade/weaken the immune system. These cells deregulate different aspects of iron homeostasis to stockpile intracellular iron. The latter is crucial in many physiological processes including cell cycle regulation, DNA synthesis in malignant cells, tumour development, metastasis and modification of the tumour microenvironment. Hence, lymphocyte function and iron metabolism are intimately related, but significance of iron imbalances in modulating the tumour microenvironment, leading to tumour progression requires further studies [115].

An overview of the effects of iron overloading and deficiency on different haematopoietic lineages has been summarised in Table 1 and Table 2, respectively.

## 5. Conclusions

Iron plays a crucial role in normal development of different haematopoietic lineages and a wide spectrum of other biological activities. Haematopoiesis is closely associated with the regulation of iron homeostasis, as too little or excess of this micro nutrient affects self-renewal and maturation of HSCs and haematopoietic lineages originating from them. Dysregulation of ROS generation during iron overloading often leads to malignant phenotypes. Intracellular iron levels and erythropoiesis are tightly linked through a network of hormones and transcription factors such as EPO, ERFE and HIF-2α which boost erythropoietic output during emergency conditions such as hypoxia, iron deficiency or stress erythropoiesis. Apart from the well-documented role of iron in erythropoiesis, another haematopoietic lineage with which intracellular iron levels have been closely associated is the megakaryocytic lineage. While certain targets and signalling pathways such as increased expression of HIF-2α, VEGF-A and reduced ERK phosphorylation have been proposed to stimulate megakaryopoiesis in iron-deficient conditions, some studies have indicated that decreased levels of iron itself may act as a stimulatory factor. Many more yet unidentified genes/transcription factors may emerge as important regulators linking iron levels to platelet biogenesis in the future.

Iron homeostasis and macrophage biology are intimately linked as the former influences polarisation of macrophages into different functional sub types, while these phagocytic cells play a prominent role in recycling iron from senescent erythrocytes and promote early erythroid maturation. Iron also influences neutrophil counts and contrasting reports have indicated decreased neutrophil apoptosis as well as neutropenia in patients affected with varying degrees of IDA. Lastly, the close association of iron with developmental regulation of the lymphoid lineage may be inferred from altered T-lymphocyte counts in IDA patients and in diseases associated with iron overloading. This may be explained by significant contributions of iron in influencing gene arrangements which affect lymphocyte maturation. Recent evidence suggests that this vital micronutrient surprisingly also affects certain epigenetic modifications, responsible for proliferation and activation of B lymphocytes. Although contradictory reports point towards a transferrin-dependent as well as independent means of iron uptake by lymphocytes, it is intimately associated with the immune system and its enhanced levels are hallmarks of several neurodegenerative diseases and tumours. Iron and haematopoiesis are intricately linked through a complex web of interactions, more of which are being discovered with ongoing research. This is an unexplored iron mine and research may shed light on future medical interventions, for treatment of diseases resulting from yet unknown cross talks between iron homeostasis and haematopoiesis.

## Figures and Tables

**Figure 1 genes-12-01270-f001:**
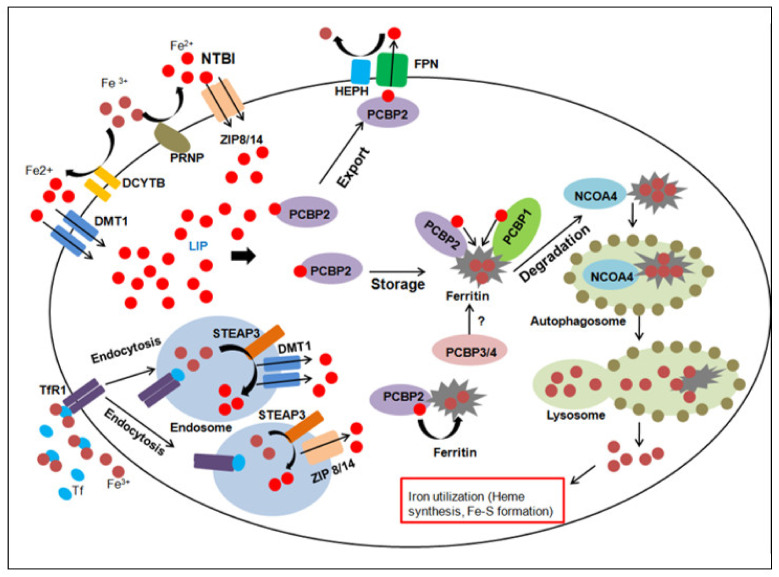
Dietary iron is principally absorbed by enterocytes of the duodenum by the divalent metal transporter 1 (DMT1) in the ferrous (Fe^2+^) state after being reduced from ferric (Fe^3+^) state by the duodenal cytochrome B (DCYTB). While transferrin-bound iron is taken up by the transferrin receptor (TfR1), non-transferrin-bound iron (NTBI) was recently discovered to be taken up by the ZRT/IRT like protein members (ZIP 8,14) in association with the ferrireductase prion protein (PRNP). Transferrin-bound iron is internalised into acidic endosomes by ‘endocytosis’, released from TfR1 and reduced to Fe^2+^ state by the epithelial antigen of prostate 3 (STEAP3). Fe^2+^ iron is released from endosomes into the cytoplasm by DMT1 and ZIP 8/14. The cytosolic labile iron pool (LIP) composed of Fe^2+^ may be delivered to the iron exporter ferroportin (FPN) by members of iron chaperones i.e., poly-(rC)-binding proteins (PCBPs). Iron is finally exported out of the cell into the circulatory system via the copper-dependent ferroxidase hephaestin (HEPH). PCBP 1 and 2 may also transport Fe^2+^ to ferritin, where it is oxidised to Fe^3+^ and stored in an intracellular non-toxic form. Iron-loaded ferritin may also undergo degradation by autophagy with the aid of the nuclear receptor coactivator 4 (NCOA4), which releases iron for cellular use, such as in synthesis of heme or iron sulphur cluster (Fe-S) formation.

**Table 1 genes-12-01270-t001:** Effects of iron overloading on the development of different haematopoietic lineages.

Affected Cells	Physiological Effects	Disease/Health Effects	Ref
HSCs	ROS production causes injury and apoptosis	Aberrant ROS levels observed in malignancies such as MDS, AML.	[38,39,41,42]
	Reduced proliferation, self renewal and apoptosis.	Iron overloading due to reduced expression of FBXL5; may contribute to disease pathology in MDS patients.	[44]
HSPCs	Elevated ROS levels induces increased expression of p38 MAPK	Observed in MDS patients, affected with iron overloading. Progression to AML may occur.	[42]
BM-MSCs	ROS production may promote cell cycle arrest and inhibit proliferation.	Differentiation into osteoblasts affected, resulting in low bone mineral density. This may account for haematopoietic niche defects and dysfunctions in iron overloading syndromes.	[48,49]
	Alternatively, increased expression of cyclin proteins and activation of MAPK signalling may promote proliferation.		[49]
	Osteoblastic commitment and matrix calcification hindered.		[49]
Erythrocytes	Impaired hepcidin synthesis results in iron overloading, in response to chronic stress/ineffective erythropoiesis.	Observed in patients suffering from β thalassemia	[65]
	Increased apoptosis of HSCs (by ROS-induced activation of p53) and reduced erythroblast differentiation due to oxidative stress, observed in BM cells of patients affected with iron overloading.	Detrimental to normal haematopoiesis, leads to anaemia.	[67]
Megakaryocytes	Iron overloading due to chronic blood transfusions affects platelet functions.	Dysfunctional platelets observed in patients affected with Diamond-Blackfan anaemia.	[75]
	Platelet counts affected in patient suffering from β thalassemia major	Pancytopenia, including thrombocytopenia observed in the patient.	[74]
Macrophages	Storage of excess intracellular iron in ferritin renders macrophages bactericidal.	These macrophages are polarised towards a pro-inflammatory phenotype.	[95]
	Macrophage iron accumulation in Fpn-deficient mice models increases ROS production and systemic inflammation.	These changes may be responsible for progression of atherosclerosis.	[97]
Neutrophils	Iron overloading severely affects their phagocytic and bactericidal activities.	Observed in β thalassemia major and patients with chronically transfused haemodialysis.	[100]
	Absence of accumulation of intracellular iron in certain diseases protects neutrophils and primes them towards phagocytosis.	Observed in mice models of HH and affected patients.	[100]
Lymphocytes	T cell counts affected in patients suffering from iron overloading syndromes	CD8+ T cells decline in HH patients while CD4+ T cells decline and CD8+ T cells increase in patients affected with β thalassemia major.	[101]
	Iron deposition has been associated with neurodegeneration, inflammation, abnormal cell proliferation and tumour metastasis.	Hallmark of neuroinflammatory diseases such as MS and multiple types of cancers, where host immune responses are aberrant/compromised.	[114,115]

**Table 2 genes-12-01270-t002:** Effects of iron deficiency on the development of different haematopoietic lineages.

Affected Cells	Physiological Effects	Disease/Health Effects	Ref
HSCs	Iron-deficient *Tfr1* knockout mice display impaired differentiation of most lineages.	Cellular iron deficiency attenuates lineage commitment and regeneration potential of HSCs, leading to postnatal lethality.	[35]
HSPCs	Loss of *Tfr1* dose not seem to be indispensable for production of HSPCs in mice fetal liver.	Cellular iron deficiency in Tfr1 knockout mice affects differentiation and regenerative capacity of HSPCs.	[35]
BM-MSCs	Iron deficiency induced by chelators like DFO protects BM-MScs from oxidative stress and increases viability.		[52]
	Iron chelators may protect BM-MSCs from toxicity by decreasing levels of ROS and inhibiting certain intracellular signaling pathways (p38 MAPK)		[48]
	Mitochondrial fragmentation in iron overloaded BM-MSCs may be reduced by iron chelators such as DFO and NAC.	Observed in MDS patients.	[48]
Erythrocytes	Hypochromic, microcytic erythrocytes typically observed in IDA patients	IDA accounts for 50% of the global burden of anemia.	[63]
	Aberrant overexpression of hepcidin despite low iron stores and serum iron levels, observed in certain anemias.	Leads to iron restricted erythropoiesis in patients affected with IRIDA and anemia of inflammation, respectively	[64]
	Differentiation and maturation of erythroid progenitors are inhibited in presence of iron deficiency, in an EPO responsive manner	Development of anemia occurs in iron deprived mice, which is reversed by application of isocitrate.	[59]
Megakaryocytes	Increase in megakaryocytic progenitor populations, greater states of ploidy proplatelet like structures reported in in vitro studies	May explain reactive thrombocytosis observed in some patients affected with mild to moderate IDA.	[73]
	Iron may influence platelet biogenesis by regulating formation of certain precursor cell types. This function is affected in severe iron deficiency.	May account for occurences of thrombocytopenia, reported in some severe IDA patients	[76]
Macrophages	Excess iron recycling and low ferritin content mobilises macrophages towards tissue repair and regeneration	These macrophages possess anti-inflammatory properties	[94]
Neutrophils	Decreased neutrophil apoptosis observed in some children affected with IDA.	May lead to autoimmune disorders/malignancies in the future.	[89]
	Contrary reports of neutropenia, observed in a patient affected with severe IDA.	May be responsible for increased risk of infections.	[99]
	IDA may be partly responsible for neutrophil hypersegmentation.		[88]
Lymphocytes T	IDA is known to affect lymphocyte counts.	Decreased CD4+ T cells and increased CD8+ counts, reported in symptomatic as well asymptomatic IDA cases.	[113]
	Lymphocytopenia reported in an adolescent IDA patient.	May pose risk for future infections.	[99]

Table 1 and Table 2, Abbreviations used—HSCs—haematopoietic stem cells; HPSCs—haematopoietic stem/progenitor cells; BM—bone marrow; BM-MSCs—bone marrow-derived mesenchymal stem cells; ROS—reactive oxygen species; MDS—myelodysplastic syndrome; AML—acute myeloid leukaemia; FBXL5—F-box/LRR-repeat protein 5; *TfR1*—transferrin receptor 1; MAPK—mitogen-activated protein kinase; DFO—deferoxamine; NAC—N-acetyl-L-cysteine; IDA—iron deficiency anaemia; IRIDA—iron-refractory iron deficiency anaemia; EPO—erythropoietin; HH—hereditary haemochromatosis; MS—multiple sclerosis; FPN—ferroportin.

## Data Availability

Not Applicable.

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
