# Peer review of "Complex Interactions in Regulation of Haematopoiesis—An Unexplored Iron Mine"

_genes, 2021, doi:10.3390/genes12081270_

Round 1

Reviewer 1 Report

The manuscript “Complex Interactions in regulation of Haematopoiesis- an unexplored Iron Mine” by R.De, KU Prakash and ES Edison is a comprehensive review regarding the significance of iron homeostasis in different aspects  of the normal development of different haematopoietic lineages. The review  is  well organized and  discussed from multiple aspects.  It would be of interest to readers of this journal. The references are appropriate and quite updated.

There are a few  suggestions to the authors:

  1. Check the spaces between the words (many of them are connected; for example L37 haemoglobinof, L55 ofthe, L65ispredominantly, Table 2 L741 dosenot...) and between the comma or the full stop and the following word (L84, L186, L189, L345, L510...). 
  2. Check the spaces when use hyphen and / (for example L2 Haematopoiesis- an, L469 heme oxygenase -1, L368 thrombocytopenia / decreased, L372 thrombocytes /platelets, L503 chelation/ dietary). 
  3. L217-L226 Thera are references 42-44-43-44. Reorder the list of references to get 42-43-44.
  4. L420 "Also, platelet" instead of "Also, Platelet".
  5. L421 Put both references together in brackets.
  6. In the reference list, references 100 and 115 are the same, there is no need for 115.   

Author Response

Comments for Reviewer-

As noted by the reviewer, some changes to the manuscript had been suggested. Please find below the following changes that have been made, in accordance with the same.

  1. Spaces between words, words and references (L37, L54, L62, L79, L92, L93, L133, L160, L178, L184, L208, L209, L294, L325, L329, L370, L379, L384, L445, L447, L470, L471, L505, L736 (Table 2), L817, L829) have been inserted.
  2. Spaces following a comma/full stop and the next word (L83, L113, L182, L258, L298, L340, L370, L471, L477, L523, L528, L593, L814) have been inserted.
  3. Spaces before/after hyphens and / have been corrected (L2, L320, L359, L363, L367, L391, L420, L423, L450, L463, L485, L497, L550, L820).
  4. Minor corrections in language & spellings have been incorporated in (L89, L240, L249, L587, L614, L688 (Table 1), L810, L811, L822).
  5. L212-L217: The references have been reordered from 42-44-43-44 to 42-43-44, as suggested.
  6. Changes have been made in L415 and L416, as suggested by Reviewers.
  7. Minor change has been made in text (L321) and L677 (Table 2).
  8. Reference 115 has been removed from the reference list, as it is same as reference no. 100, changes relevant to the same have been incorporated in the text in appropriate places.
  9. Changes in references have been made (L625, L634, L720 (Table 1) and L795 (Table 2).

Reviewer 2 Report

The reviewed paper present many aspects of iron contributing to maintaining and proliferation of stem cells, maintenance of mesenchymal stem cells and in immunology. The authors describe detailed influence of iron on myelopoiesis. Only one my remark is on the role of the rag2 gene. They went through literature meticulously on the theme as they present in the title and show us all aspects in presented fields nowadays.

The part 562-570 about Rag2 on in iron is not understandable, what has common with it?

Author Response

Comments for Reviewer –

  • As noted by the Reviewer, the paragraph detailing the significance of mice lacking the Rag2 gene on iron does not seem to have a direct correlation and requires further explanation [L555-L563].
  • Although, no direct correlation of the absence of Rag2 gene on iron absorption by mice exists, the authors in this paragraph wanted to emphasize that mice carrying a homozygous deletion of Tfr gene on a Rag2 null background (Tfr-/-, Rag2-/-) displayed impaired B lymphocyte differentiation.
  • Antibody class switching seems to be affected in these mice, resulting in only a few chimeric mice producing IgM on surfaces of B lymphocytes, while other antibody classes were not produced. Thus, iron uptake through Tfr appears to be significant for early lymphoid cell maturation.